# Sorption Behaviors of Light Lanthanides(III) (La(III), Ce(III), Pr(III), Nd(III)) and Cr(III) Using Nitrolite

**DOI:** 10.3390/ma13102256

**Published:** 2020-05-14

**Authors:** Grzegorz Wójcik

**Affiliations:** Department of Inorganic Chemistry, Institute of Chemical Sciences, Faculty of Chemistry, Maria Curie-Skłodowska University, 20-031 Lublin, Poland; g.wojcik@poczta.umcs.lublin.pl; Tel.: +48-537-5732

**Keywords:** lanthanides, chromium, kinetics, removal, sorption, Nitrolite

## Abstract

The sorption of light lanthanides(III) (La(III), Ce(III), Pr(III), Nd(III)) and chromium(III) ions from acidic solutions on Nitrolite was studied at varying ions concentrations, pH, contact time and temperatures. The sorption capacity of lanthanides(III) and chromium(III) ions were examined in the ranges 2–9 and 2–5, respectively. The adsorption capacities of all metals are increase with the increasing pH (up to initial pH 9), despite the potential precipitation of metals at higher pH values. Therefore, an initial pH 9 of lanthanides gives the highest adsorption capacities. The kinetics of sorption chromium(III) and light lanthanides(III) were investigated. The experimental data were analyzed using the pseudo-first-order, pseudo-second-order forms, Elovich, and intra-particle diffusion models. The sorption kinetics of investigated ions was described by pseudo-second-order model the best. The results indicate the endothermic process of Cr(III), La(III), Ce(III), Pr(III) and Nd(III) ions sorption. The sorption capacities of La(III) 4.77 mg/g, Ce(III) 4.45 mg/g, Pr(III) 4.30 mg/g, Nd(III) 4.13 mg/g and Cr(III) 2.39 mg/g were calculated from the Langmiur model, which describes adsorption better than Freundlich and Dubinin–Radushkevich.

## 1. Introduction

Every day, millions of people around the world use electronic devices. In rich countries, wealthy people buy new devices without need, because they want to have the latest model. Electronic devices such as mobile phones contain many metals in their structure. There are three groups of metals: (a) basic metals: Cu, Al, Sn, Ni, Zn, Cr, Pb, Si, (b) precious metals: Ag, Au, (c) rare earth elements: La, Ce [1]. The basic metals group includes chromium(VI) which has mutagenic effects while chromium(III) is not toxic [2]. Therefore proper storage and treatment of e-wastes (electronic wastes) is necessary, because basic metals like: Cd, Ni, Zn, Cr, Pb can cause environmental contamination [3]. Gold and other precious metals are valuable, so they should be recovered. For this purpose microorganisms bioleaching can be carried out from the printed circuit board [4]. In addition to the precious metals, rare earth elements REEs are very valuable. Without REEs, there would be no mobile phones, lasers, electric cars and many other electronic devices. Therefore REEs should be recovered, purified and reused in electronics [5]. Lanthanide ions can be removed from acidic solutions by the solvent extraction process. Cyanex 923 is a solvating extractant with suitable properties for extracting chloride complexes of lanthanides [6]. Lanthanides can be extracted from different acids, such as: phosphoric, nitric or sulfuric ones. The extraction of lanthanides from hydrochloric, phosphoric and sulfuric acids declines with the increasing acidity [7]. Tributyl phosphate (TBP) is a very popular extractant for Ce(VI) ions. TBP is selective only for Ce(IV), considering all lanthanides(III) ions. Moreover, TBP can be immobilized in the polymeric matrix like Amberlite XAD 16 [8]. TBP was used for the solvent extraction of cerium(III) from the phosphoric acid solution [9]. Besides the solvent extraction, ion exchange can be used for lanthanides sorption. The weak base anion exchanger Purolite S 910 was used for lanthanides removal. The best results were obtained at pH 4.38 in the acetic acid media [10]. Chromium(VI) ions can be selectively extracted by Aliquate 336. This is a quaternary ammonium extractant, which extracts Cr(VI) ions from acidic media effectively. Above pH 7 the extraction efficiency decreases [11]. The activated carbon was tested with respect to chromium(III) and (VI) adsorption; the degree of removal was higher at pH 6 than at pH 9 [12]. The two anion exchangers: strong-base Lewatit MP 610 and weak-base Lewatit MP 62, were investigated for chromium(VI) ions removal. It was observed that the maximum adsorption capacity for Lewatit MP 62 and Lewatit M 610 was achieved at pH 5.0 [13]. As can be seen from the literature review, both the solvent extraction (SX) and ion exchange (IX) methods are common and useful for chromium and lanthanide ions removal from solutions. However, these methods have some disadvantages: solvent extraction needs toxic extractants and organic solutions for its dissolution, but ion exchangers are expensive. For this reason, natural sorbents, which are cheap and easily accessible, can be used for chromium and lanthanide ions removal. Nitrolite is a mineral sorbent from the zeolites group. The high aluminum concentration, as well as the presence of the main exchangeable cations, are directly related to the ion exchange capacity of Nitrolite. Nitrolite is a commercial product from the Purolite Company. The Nitrolite was used for the removal of PO_4_^3−^, NO_3_^−^ and NH_4_^+^ ions from waste waters [14,15]. This sorbent was not examined for lanthanides removal, according to the Web of Science data base.

The main goal of this paper was the investigation of the applicability of Nitrolite for chromium(III) and lanthanides(III) ions removal from the solutions in the pH range 2–9.

## 2. Materials and Methods

### 2.1. Materials

Nitrolite was obtained from Purolite International, Ltd.,UK. Nitrolite was washed prior to the experiments in order to remove the surface dust. The physicochemical characteristics are presented in Table 1 [16].

Then, the sorbent was washed with demineralized water and dried at room temperature. The La(III), Ce(III), Pr(III), Nd(III), Cr(III) solutions were prepared form ROMIL PrimAg^®^ Mono-Component Reference Solutions. The metal ion solutions were in the nitrate form, being prepared from nitrate salts. The water used for investigation was HPLC grade and was obtained from the Hydrolab HLP 10 system. Its conductivity was 0.1 uS/cm. The pH values were adjusted by using sodium hydroxide and nitric acid solutions.

### 2.2. Instrumentation

Inductively coupled plasma spectrometry (Varian 720 ES ICP-OES) was used for all metal ions analysis. Chromium (VI) ions concentration was determined by the UV-Vis method with diphenylcarbazide. The pH value was measured using a glass electrode connected to the Elmetron CP-401 pH meter. The Nitrolite samples were shaken using the laboratory shaker Elpin+ type 358, Lubawa, Poland, at the amplitude 8 and speed 150 c.p.m. The FTIR-ATR spectra were obtained, applying the Agilent Technologies Cary 630 spectrometer (Agilent Technologies, Santa Clara, CA, USA). The samples were crushed in a mortar just before the measurements.

### 2.3. Experimental Methods

In the next experiment, 1 g of Nitrolite was equilibrated with 50 mL of La(III), Ce(III), Pr(III), Nd(III) Cr(III), at the solution concentration 100 mg/L, in the shaker bath at 293 K. The samples were shaken in the contact time range 1–360 min. The sorption experiment was performed in the pH range 2–5. The ion exchange capacity, *q_e_* (mg/g), was calculated using Equation (1).
(1)$$q_{e}=\frac{C_{0}-C_{e}}{m}\times \frac{V}{1000}$$
where: *C*_0_ and *C_e_* are the initial and equilibrium concentrations of ions, respectively [mg/L], *V* is the solution volume [L], m is the mass of Nitrolite [g].

In order to determine the isotherm 1 g of Nitrolite was equilibrated at 293 K with 50 mL of La(III), Ce(III), Pr(III), Nd(III) Cr(III) solutions in the concentration range 100–1000 mg/L in the shaker bath for 24 h. In the temperature experiment, 1 g of Nitrolite was contacted with 50 mL of the right ion at three different temperatures; 293, 303, 333 K. The desorption experiments were performed by contacting 0.5 g of Nitrolite with the adsorbed individual metal ions with 25 cm^3^ of 1 M ammonium acetate in the 24 h contact time and shaken at the amplitude 8 and speed 150 c.p.m.

The removal of light lanthanides(III) and chromium(III) from chromium(VI) ions solutions was done at pH value 3.5. 1 g of Nitrolite was equilibrated with 50 mL of the solution containing La(III), Ce(III), Pr(III), Nd(III), Cr(III) and chromium(VI) at concentration of 100 mg/L of each metal ions.

## 3. Results and Discussion

### 3.1. Effects of Contact Time and pH Value

The influence of the contact time on the sorption of light lanthanides(III) in the acidic media is presented in Figure 1a–e. For all investigated lanthanides(III), the sorption capacity depends on the pH values. The sorption capacity increases with the increasing pH values. The difference in the sorption capacity is lower for Pr(III) and Nd(III) than for La(III) and Ce(III), in the pH range 3.5–5. Light lanthanides differ in the size of the ions i.e., La(III) 1.15 Å, Ce(III) 1.11 Å, Pr(III) 1.06 Å, Nd(III) 1.08 Å. Moreover, La(III) is hydrated to a smaller extent than Ce(III), Pr(III) and Nd(III), which could be the reason for some differences in sorption. However, the light lanthanides are so similar, that the differences in sorption, thus the selectivity of Nitrolite towards them, is low. At pH 2, the sorption capacity is the lowest for all studied light lanthanides (La(III), Ce(III), Pr(III), Nd(III)). This phenomenon can be explained by the competitive reactions of metal ions and hydrogen cations on the sorption sites in Nitrolite. At pH 2, the concentration of hydrogen cations is much higher than in the pH range 3.5–5. Moreover, when Nitrolite was contacted with the demineralized water at pH 2 and 5, the concertation of Na, K, and Ca was higher at pH 2 than at pH 5. This indicates the ion exchange process of hydrogen cations. The sorption capacities of chromium(III) ions increase with increasing pH and were respectively: 0.6 mg/g at pH 2, 1.9 mg/g at pH 3.5 and 2.4 mg/g at pH 5. During the ion exchange reaction of lanthanides and chromium ions, the initial pH values were changed and those of the equilibrium pH are presented in Figure 2. In the initial pH range 2–5 the values of equilibrium pH are higher for lanthanides, but in that 7–9 the equilibrium pH values decrease. A similar observation was made for chromium(III) at the initial pH 5. To understand the sorption process of lanthanides and chromium ions better, the chart of equilibrium pH vs. the sorption capacity was prepared and is presented in Figure 3.

As can be seen from Figure 3, the sorption capacities of lanthanides and chromium increase for the equilibrium pH range 2–7. The maximum capacities for lanthanides were observed at the equilibrium pH about 6.8–6.9. The higher values of pH can promote the precipitation process on Nitrolite. The values of lanthanides and chromium pH were calculated using computer program Medusa. Lanthanum(III) in the pH range 6–9 exist partly as La(OH)_2_^+^, and at pH 7.8, its precipitation can start. Ce(III) can form Ce(OH)_2_^+^ and Ce(OH)^2+^, in the pH range 6–8 and starts precipitation at pH 7.6. Similarly, Pr(OH)_2_^+^ can exist in the pH range 5.6–8.4, and its precipitation starts at pH 7.35. Neodymium in the pH range 5.5–8 can exist as Nd(OH)_2_^+^ and begins to precipitate at pH 6.9, which is the lowest value compared to those for La(III), Ce(III), Pr(III). Similar behaviour was observed for chromium(III) ions in the equilibrium pH range 3.6–4.2. Chromium can exist as Cr(OH)_2_^+^, Cr_2_(OH)_2_^4+^, Cr_3_(OH)_4_^5+^, in the pH range 2.8–4.2, and its precipitation can start at pH 4.3. The presence of hydroxyl forms of chromium and lanthanides can also affect the capacity increase and results in lower equilibrium pH.

### 3.2. Kinetic Models

Several kinetic models were used to describe the sorption of lanthanides(III) and chromium(III) ions on Nitrolite. The most common are: pseudo-first-order, pseudo-second-order, Elovich and intra-particle kinetic models. The kinetic models are presented in Table 2 [17,18].

From the equations presented in Table 2, the kinetic parameters were calculated and the results are given in Table 3. For all lanthanides(III) and chromium(III), the parameters were calculated using the pseudo-first-order PFO, pseudo-second-order PSO1, PSO2, PRO3, PSO4, Elovich and intra-particle diffusion kinetic models. For the investigated three positive ions, low values of the determination coefficient were obtained, based on the intra particle diffusion model. This indicates that the sorption of three positive ions is not limited by intra-particle diffusion. The lowest values of the determination coefficient are obtained for the PSO3 and PSO4 models for all ions and the R^2^ values were in the range 0.2–0.3. The best results were found considering the pseudo-second order model PSO1. The determination coefficients for the PSO1 model are high and close to 0.99. Moreover, the calculated sorption capacities are consistent with those presented in Figure 1. The calculated sorption capacities increase with the increasing pH values from 2 to 5.

### 3.3. Sorption Isotherm

The equilibrium data were analyzed by the most commonly used Langmuir [19], Freundlich [20], and Dubinin–Radushkevich (D-R) [21] isotherm models.

The Langmuir isotherm was calculated based on Equation (2):(2)$$\frac{C_{e}}{q_{e}}=\frac{1}{b\times Q_{0}}+\frac{C_{e}}{Q_{0}}$$
where, *q_e_* is the amount of adsorbate in the adsorbent (mg/g); *C_e_* is the equilibrium concentration (mg/L); *b* is the Langmuir isotherm constant (L/mg); *Q*_0_ is the maximum monolayer coverage capacity (mg/g).

The values of the determination coefficient (R^2^) in the Langmuir isotherm for lanthanides(III) and chromium(III) are about 0.99 and show that this model describes the experimental data better than the Freundlich and D-R models. The capacities for lanthanides(III) are from 4.7 mg/g for La(III) to 4.13 mg/g for Nd(III) at pH 5. The values of sorption capacity *Q*_0_ for chromium(III) are lower, being 2.36 mg/g at pH 3.5. Chromium(III) ions can start precipitation at pH 4.2, but lanthanides cannot precipitate, because they occur as three positive cations. Moreover, the Langmuir isotherm allows one to estimate the *R_L_* parameter. It can be calculated using Equation (3).
(3)$$R_{L}=\frac{1}{1+b\times C_{o}}$$
where *C*_0_ is the initial concentration of metal ions in solution (mg/L)

The *R_L_* parameters calculated from the Langmuir isotherm for lanthanides(III) and chromium(III) are presented in Table 4. As can be seen, the values of *R_L_* are in the range 0.0254–0.0437. This points to the favorable sorption of lanthanides(III) and chromium(III) on Nitrolite.

The Freundlich isotherm was calculated according to Equation (4):(4)$$logq_{e}=logK_{F}+\frac{1}{n}logC_{e}$$
where *q_e_* is the amount of adsorbate in the adsorbent (mg/g); *K_F_* is the characteristic constant related to the adsorption capacity (L/g); *n* is the adsorption intensity; *C_e_* is the equilibrium concentration (mg/L).

The values of determination coefficient for the Freundlich isotherm are lower than those for the Langmuir one (Table 4). This indicates that the Freundlich model does not describe the sorption of lanthanides(III) and chromium(III) as well as the Langmuir one. This is due to the fact that lanthanides(III) are sorbed by the ion-exchange mechanism.

The linear form of the Dubinin–Radushkevich (D-R) model [20] is presented in Equation (5)
(5)$$lnq_{e}=lnq_{m}-\beta \times \varepsilon^{2}$$
where *q_m_* is the maximum of chromium(III) and lanthanide(III) ions in Nitrolite (mol/g); β is related to the adsorption energy (mol^2^/kJ^2^), ε is the Polanyi potential, which can be calculated from Equation (6)
(6)$$\varepsilon =RTln\left[1+\frac{1}{C_{e}}\right]$$

Taking into account the β parameter, the D–R model allows one to calculate adsorption free energy E (kJ/mol)
(7)$$E=\frac{1}{\sqrt{2\beta }}$$

The results of the Dubinin–Radushkevich parameter calculated for lantanides(III) and chromium(III) are presented in Table 4. The values of the determination coefficients are lower for the D-R isotherm than those for the Langmuir isotherm. The Dubinin–Radushkevich isotherm is very useful because it allows one to calculate the adsorption energy E, which is from 15.49 kJ/mol for Nd(III) to 15.99 kJ/mol for La(III) and 14.87 kJ/mol for Cr(III) ions. The calculated values are in the range of 8–16 kJ/mol, which indicates the chemisorption mechanism in the case of lanthanides(III) and chromium(III) ions sorption.

The values of equilibrium pH decreased with the increasing lanthanides concentration in the initial solution. Lanthanides were contacted with Nitrolite at the initial pH 5. After the 24 h contact time, the equilibrium pH was from 5.85 at 100 mg/L to 5.67 at 1000 mg/L for La(III). Similar pH values were observed for Ce(III), Pr(III) and Nd(III). This may be due to lanthanide ions complexation by the hydroxyl group present on the Nitrolite surface. A similar behaviour was observed during Nd(III) sorption on clinoptylolite [22]. This can be presented by the equations:
3HO-Nitrolite + La^3+^ ⇄ La(O-Nitrolite)_3_ + 3H^+^(8)
3HO-Nitrolite + Ce^3+^ ⇄ Ce(O-Nitrolite)_3_ + 3H^+^(9)
3HO-Nitrolite + Pr^3+^ ⇄ Pr(O-Nitrolite)_3_ + 3H^+^(10)
3HO-Nitrolite + Nd^3+^ ⇄ Nd(O-Nitrolite)_3_ + 3H^+^(11)

Taking into account the pH changes, the main mechanism of sorption is cation exchange, but at a higher concentration of lanthanides(III) in solution the proton exchange mechanism takes place.

The equilibrium pH values for chromium(III) ions were changed from 3.64 at 100 mg/L to 3.42 at 1000 mg/L. The isotherms at the initial pH 5 for lanthanides(III) and 3.5 for chromium(III) are presented in Figure 4.

### 3.4. Thermodynamic Parameters

The thermodynamic parameters like: Δ*G*° Gibbs free energy (kJ/mol), Δ*H*° enthalpy (kJ/mol) and Δ*S*° entropy (kJ/K·mol) were calculated for Cr(III), La(III), Ce(III), Pr(III) and Nd(III) ions. They allow one to predict the nature of investigated ions adsorption on Nitrolite. They were determined from the equations:(12)$$\Delta G^{{}^{\circ}}=-RTln\left(K_{D}\right)$$
(13)$$\ln\left(K_{D}\right)=-\frac{\Delta H^{{}^{\circ}}}{RT}+\frac{\Delta S^{{}^{\circ}}}{R}$$

The values of Δ*H*° enthalpy (kJ/mol) and Δ*S*° entropy (kJ/K·mol) were obtained from the slope and intercept of the plots ln(*K_D_*) vs. 1/T; *K_D_* is the distribution coefficient (L/g), R is the universal gas constant (8.314 × 10^−3^ kJ/K·mol). The values of the thermodynamic parameters are presented in Table 5. For the chromium(III) ions sorption on Nitrolite, the positive Δ*H*° value indicates endothermic interactions at 293–323 K. The insignificantly positive Δ*S*° value indicates that randomness increases at the solid-solution interface during the chromium(III) sorption. The positive Δ*G*° values show the existence of an energy barrier and that the reaction is a non-spontaneous process. The values of free enthalpy are lower for light lanthanides(III) than for chromium(III) ions. This indicates that light lanthanides(III) are sorbed at a lower energy barrier than chromium(III) ions. The values of entropy are positive for lanthanides(III) and chromium(III) ions. According to the ion exchange reaction, the entropy increases because two positive calcium ions are exchanged into three positive lanthanides(III) and chromium(III) ions. Summing up, the sorption of investigated three positive ions proceeds more efficiently with the increasing temperature of the solution.

### 3.5. Desorption of Metal Ions

The desorption of light lanthanides(III) and chromium(III) was conducted with the 1M ammonium acetate solution at pH 5. The ammonium acetate solution desorbs the ions, which were sorbed on Nitrolite according to the ion-exchange mechanism [23]. If the amounts of desorbed ions are equal to those sorbed; this indicates that the ion exchange is a dominant process. Desorption of lanthanides and chromium ions proceeds efficiently in 1M ammonium acetate. All investigated lanthanides are desorbed, with the efficiency being higher than 95%. The desorption values for chromium(III) ions are about 99%. This indicates that lanthanides(III) and chromium(III) ions are exchanged during the sorption process on Nitrolite.

### 3.6. Removal of Light Lanthanides(III) and Chromium(III) from Chromium(VI) Ions Solutions

The removal of lanthanides(III) and chromium(III) from chromium(VI) ions solutions is possible, with very good results. Nitrolite surface does not contain positively charged groups and the chromium (VI) anions are not sorbed. The sorption experiment at pH 3.5 proved that chromate sorption is lower than 10^−2^ mg/g, which is a very low value. During the sorption process of lanthanides(III) and chromium(III) from chromium(VI) solution at pH 3.5, chromium(VI) ions were not sorbed on Nitrolite. After the sorption process, lanthanides(III) and chromium(III) ions were desorbed in 1 M ammonium acetate. This procedure allows to recover lanthanides(III) and chromium(III) ions from the chromium(VI) solution. This method was tested using the 100 mg/L of chromium(VI) and 100 mg/L of other ions solution. Both chromium(VI) and chromium(III) ions concentration were determined. The speciation analyses (simultaneous determination of both chromium (III and VI) ions) of solution show that chromium(VI) ions are not sorbed, while chromium(III) was removed from the solution at pH 3.5, with the removal efficiency 48%. Then chromium(III) ions were desorbed with 99% efficiency. This confirms that Nitrolite is able to remove three positive ions from the chromium (VI) solution.

### 3.7. ATR-FTIR Spectra

The ATR-FTIR spectra were obtained for Nitrolite after the metal ions sorption. Light lanthanides(III) were sorbed from the solution at pH 5, while chromium(III) ions were sorbed at pH 3.5. The spectra are presented in Figure 5. The most important region of the spectra is 1300–500 cm^−1^. Here, the changes in the spectra are related with sorption of lanthanides(III) and chromium(III). The band at 598.3 cm^−1^ is shifted to 600.2 cm^−1^, after the sorption of three positive ions. The band at 600.2 cm^−1^ is attributed to the pseudo-lattice vibrations in natural zeolite [24]. The bands at 671.8 cm^−1^, 728.8 cm^−1^, and 792.25 cm^−1^ can be assigned to symmetrical stretching vibrations of T-O-T (T-Si, Al) [25].

The band at 1015.9 cm^−1^ is shifted to 1019.67 cm^−1^ after the sorption of three positive ions and is related to the internal linkages between SiO_4_ and AlO_4_ tetrahedra, (Si-O) and (Al-O) asymmetric stretching vibrations [26]. Similar external vibrations between SiO_4_ and AlO_4_ tetrahedra can be observed at 1204.22 cm^−1^ [27]. The maximum at 1625 cm^−1^ corresponds to the bending vibrations of water in Nitrolite. The border band between 3600–3400 cm^−1^ results from the stretching vibrations of the OH groups [28].

### 3.8. Comparison with the Other Natural Sorbents

The Transcarpathian clinoptilolite was investigated for Pr(III) ions sorption. The highest sorption capacity value 45.7 mg/g of clinoptilolite for Pr(III) was obtained in the weakly alkaline solution at pH 9. Its sorption capacity was higher than for Nitrolite, because at pH 9, Pr(III) can exist as Pr(OH)_3_ [29]. The natural sorbent (transcarpathian clinoptilolite (KL)) was used for the immobilization of selected heavy metals in the sewage sludge. The sorption capacity for Cr(III) ion was 1.12 mg/g, being lower than for Nitrolite, which was 2.36 mg/g [30]. The natural sorbent Kaolinite was used for the removal of Nd ions. At an initial concentration of 42.3 µmol/L of neodymium, the capacity was 0.576 mg/g and was lower than for Nitrolite [31]. Smectite was used for the sorption of lanthanides. This natural sorbent is able to adsorb light lanthanides similarly to Nitrolite in the pH range 3–8 [32]. The natural sorbents can be modified to improve the sorption capacity. The Montmorillonite was modified by hexadecyl trimethyl ammonium bromide (HDTMA) and sodium diethyl dithiocarbamate (DDTC). The maximum of sorption capacity for La(III) was 0.178 mmol/g and was higher than for Nitrolite [33].

## 4. Conclusions

The research carried out and discussed allows the following conclusions to be drawn.

(a) Nitrolite is a natural sorbent allowing one to sorb Cr(III), La(III), Ce(III), Pr(III) and Nd(III) ions.

(b) Sorption kinetics of Cr(III), La(III), Ce(III), Pr(III) and Nd(III) ions can be described by the PSO1 model.

(c) The values of sorption energy calculated from the D-R isotherms are in the range 8–16 kJ/mol and indicate the chemisorption of Cr(III), La(III), Ce(III), Pr(III) and Nd(III) ions on Nitrolite.

(d) The removal of Cr(III), La(III), Ce(III), Pr(III) and Nd(III) from the chromium(VI) solutions was obtained on Nitrolite.

(e) Moreover, 1M ammonium acetate is able to desorb Cr(III), La(III), Ce(III), Pr(III) and Nd(III) ions from Nitrolite

(f) The sorption of the investigated three positive ions proceeds more efficiently with the increasing temperature of solution.

(g) The mechanism of the sorption of Cr(III), La(III), Ce(III), Pr(III) and Nd(III) ions is ion exchanging.

(h) The Nitrolite can be modified to improve its sorption capacity for lanthanide ions in the future.

## Figures and Tables

**Figure 1 materials-13-02256-f001:**
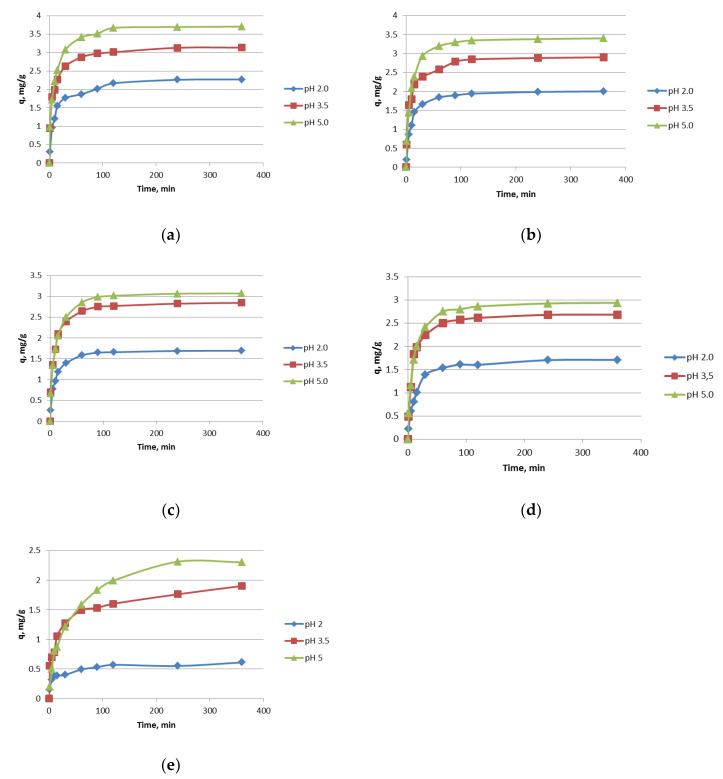
The effects of the contact time on the sorption capacity of ions in the pH range 2–5, (**a**) La(III), (**b**) Ce(III), (**c**) Pr(III), (**d**) Nd(III), (**e**) Cr(III).

**Figure 2 materials-13-02256-f002:**
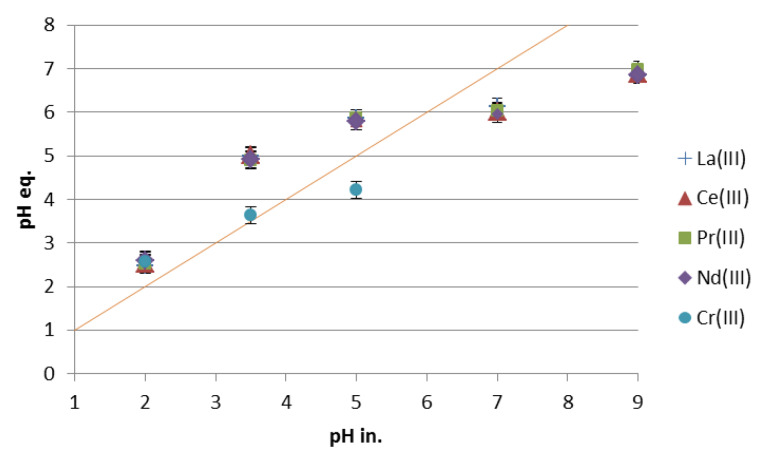
The effect of initial pH on equilibrium pH values.

**Figure 3 materials-13-02256-f003:**
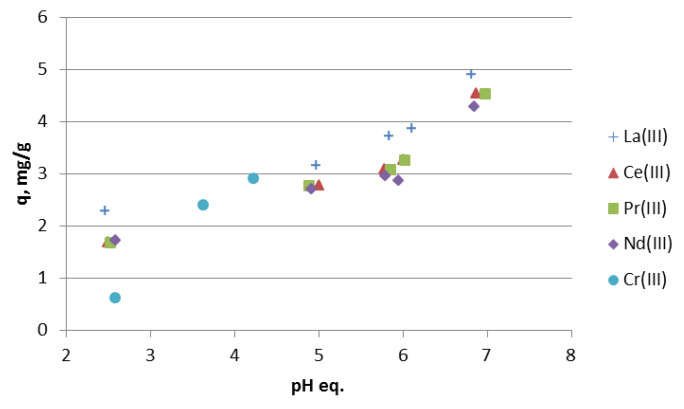
The effect of equilibrium pH on the sorption capacities of lanthanides and chromium ions. The initial concentration of metal ions was 100 mg/L.

**Figure 4 materials-13-02256-f004:**
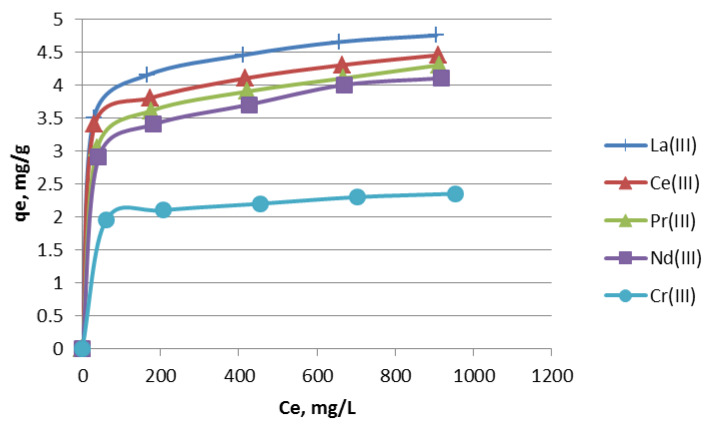
The isotherms of La(III), Ce(III), Pr(III), Nd(III) and Cr(III) ions sorption on Nitrolite.

**Figure 5 materials-13-02256-f005:**
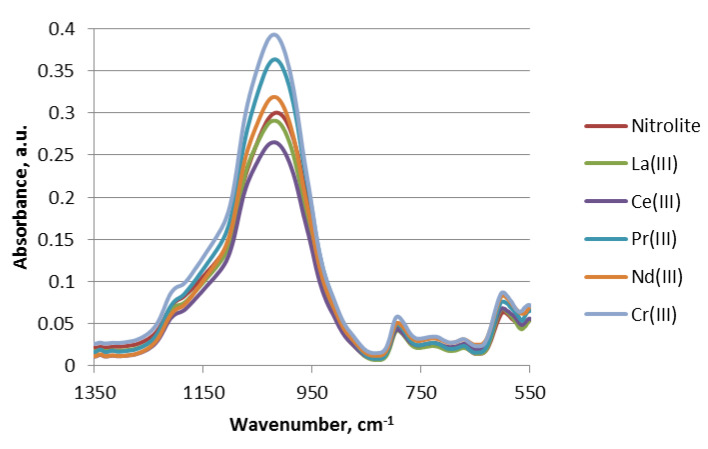
The ATR - FTIR spectra for Cr(III), La(III), Ce(III), Pr(III) and Nd(III) ions on Nitrolite.

**Table 1 materials-13-02256-t001:** Characteristics of Nitrolite.

Composition	(K,Na,1/2Ca)_2_ ·Al_2_O_3_·10SiO_2_·H_2_O
Total Capacity	0.7 eq L^−1^ (NH_4_^+^ ions)
Bead size	0.63–1.4 mm
Max temp. range	1023 K

**Table 2 materials-13-02256-t002:** The linear forms of kinetic models.

Name	Linear Form
Pseudo-first-order (PFO)	$\ln\left(q_{e}-q_{t}\right)=lnq_{e}-k_{1}\times t$
Pseudo-second-order 1 (PSO1)	$\frac{t}{q_{t}}=\frac{1}{k_{2}\cdot q_{e}^{2}}+\frac{1}{q_{e}}\times t$
Pseudo-second-order 2 (PSO2)	$\frac{1}{q_{t}}=\left(\frac{1}{k_{2}\cdot q_{e}^{2}}\right)\times \frac{1}{t}+\frac{1}{q_{e}}$
Pseudo-second-order 3 (PSO3)	$q_{t}=q_{e}-\left(\frac{1}{k_{2}\cdot q_{e}}\right)\times \frac{q_{t}}{t}$
Pseudo-second-order 4 (PSO4)	$\frac{q_{t}}{t}=k_{2}\times q_{e}^{2}-k_{2}\times q_{e}\times q_{t}$
Elovich	$q_{e}=\frac{1}{\beta }\times ln\left(\alpha \times \beta \right)+\frac{1}{\beta }\times lnt$
Intra-particle diffusion	$q_{t}=k_{ip}\times t^{0.5}+C$

**Table 3 materials-13-02256-t003:** The kinetic parameters for La(III), Ce(III), Pr(III), Nd(III), Cr(III) ions sorption.

		**PFO**			**PSO1**		
**Metal Ions**	**pH**	**k_1_ (1/min)**	**q_e_ (mg/g)**	**R^2^**	**k_2_ (g/mg** **·** **min)**	**q_e_ (mg/g)**	**R^2^**
La(III)	2	0.0131	1.3733	0.9114	0.0537	2.3144	0.9989
La(III)	5	0.0154	2.2280	0.8485	0.0535	3.7597	0.9996
Ce(III)	2	0.0141	0.6544	0.8727	0.0790	2.0347	0.9996
Ce(III)	5	0.0138	1.5482	0.8080	0.0617	3.4466	0.9997
Pr(III)	2	0.0161	0.3756	0.8658	0.1120	1.7224	0.9997
Pr(III)	5	0.0130	1.4055	0.7887	0.0609	3.1213	0.9996
Nd(III)	2	0.0145	0.6742	0.8983	0.0720	1.7481	0.9993
Nd(III)	5	0.0105	1.4310	0.7559	0.0604	2.9862	0.9997
Cr(III)	2	0.0093	0.0669	0.8514	0.2048	0.6031	0.9957
Cr(III)	5	0.00935	3.0528	0.9156	0.01665	2.46105	0.99626
		**Elovich**			**Intra-particle Diffusion**		
	**pH**	**α (g/mg·min)**	**β (mg/g)**	**R^2^**	**k (mg/g·min^0.5^)**	**C (mg/g)**	**R^2^**
La(III)	2	1.1954	2.9111	0.9663	0.1095	0.5746	0.7815
La(III)	5	3.4280	1.9769	0.9525	0.1686	1.1647	0.6784
Ce(III)	2	1.1061	3.1860	0.9286	0.0969	0.5372	0.7325
Ce(III)	5	2.5139	2.0221	0.9273	0.1584	1.0301	0.6545
Pr(III)	2	1.1945	3.9270	0.9375	0.0806	0.4867	0.7422
Pr(III)	5	2.1554	2.2328	0.9480	0.1450	0.8882	0.6875
Nd(III)	2	0.7362	3.5966	0.9504	0.0867	0.3825	0.7978
Nd(III)	5	1.8693	2.2813	0.9405	0.1399	0.8297	0.6811
Cr(III)	2	0.8688	13.4576	0.9646	0.0259	0.1900	0.7247
Cr(III)	5	0.4297	2.48194	0.9687	0.12051	0.2753	0.89843

**Table 4 materials-13-02256-t004:** The Langmuir, Freundlich and the D–R isotherm model constants and the determination coefficients for Cr(III), La(III), Ce(III), Pr(III) and Nd(III) ions sorption on Nitrolite.

Metal Ions	Langmuir				Freundlich			Dubinin–Radushkevich			
	Q_0_	b	R_L_	R^2^	K_f_	n	R^2^	β	q_m_	E	R^2^
La(III)	4.77	0.0698	0.0278	0.9990	1.1952	4.49	0.9066	1.96 × 10^−9^	0.000246	15.99	0.9575
Ce(III)	4.45	0.0603	0.0321	0.9982	1.1849	4.73	0.9027	2.03 × 10^−9^	0.000232	15.71	0.9703
Pr(III)	4.30	0.0478	0.0401	0.9975	1.1254	4.76	0.9439	2.05 × 10^−9^	0.000223	15.63	0.9776
Nd(III)	4.13	0.0438	0.0437	0.9974	1.1042	4.88	0.9559	2.08 × 10^−9^	0.000212	15.49	0.9882
Cr(III)	2.39	0.0768	0.0254	0.9989	1.0487	8.05	0.9647	2.26 × 10^−9^	0.000289	14.87	0.9485

**Table 5 materials-13-02256-t005:** The values of thermodynamic parameters for Cr(III), La(III), Ce(III), Pr(III) and Nd(III) ions.

Metal Ions	ΔH° (kJ/mol)	ΔS° (kJ/K·mol)	ΔG° (kJ/mol)		
			293 K	303 K	323 K
La(III)	10.224	0.0154	5.67	5.52	5.04
Ce(III)	10.867	0.0154	6.30	6.18	5.67
Pr(III)	13.305	0.0200	7.39	7.16	6.58
Nd(III)	11.217	0.0128	7.57	6.94	7.09
Cr(III)	17.580	0.0264	9.72	9.63	8.62

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
