# Peer review of "Sorption Behaviors of Light Lanthanides(III) (La(III), Ce(III), Pr(III), Nd(III)) and Cr(III) Using Nitrolite"

_materials, 2020, doi:10.3390/ma13102256_

Round 1
Reviewer 1 Report
This is an interesting paper on the sorption of La and Cr ions from acidic solutions on Nitrolite at various ions concentrations, pH, contact time and temperatures. Ti improve the paper I would suggest that the art work must become a little more clear.
Figure 1 and 4 need a line to connect isotherms.
Figure 2 need error bars to be introduced.
Figure 5 seems to be unnecessary or presented by a histogram rather than a bar chart.
Conclusion must be extended.
Author Response
Response to Reviewer 1 Comments
Thank you very much for kind revision. All comments were included in the revised paper. It allowed to improve significantly the paper.
Comment 1
Figure 1 and 4 need a line to connect isotherms.
Response 1
The lines have been added to the Figure 1 and 4
Comment 2
Figure 2 need error bars to be introduced.
Response 2
The error bars have been added to Figure 2
Comment 3
Figure 5 seems to be unnecessary or presented by a histogram rather than a bar chart.
Response 3
Figure 5 has been deleted.
Comment 4
Conclusion must be extended.
Response 4
The conclusion have been extended.
f) The sorption of investigated three positive ions proceeds more efficiently with the increasing temperature of solutions
g) The mechanism of sorption of Cr(III), La(III), Ce(III), Pr(III) and Nd(III) ions is ion exchanging.
h) The Nitrolite can be modified to improve its sorption capacity for lanthanide ions in the future.

Reviewer 2 Report
This script describes the adsorption ability of nitrolite for light lanthanides(III) and chromium(III), and discussion for the applicability of nitrolite to remove these elements from wastewater. It is an interesting script, but it is not written properly. There are some points, which require major revision and need to be clarified in the revised text. The points are described below.
- Title: This is very ambiguous. For example, “Sorption behaviors of light lanthanides (III) (La(III), Ce(III), Pr(III), Nd(III)) and Cr(III) using Nitrolite” is better to understand this paper.
- Abstract 9: “light lanthanides (III)”→”light lanthanides (III) (La(III), Ce(III), Pr(III), Nd(III))”. i
- Abstract 10: “The investigated ins were ,,,(ICP-OES).” delete.
- Abstract: You should add the isotherm results and indicate which isotherm models and kinetics models are the best.
- Introduction: There are shortage of the information for Nitrolite. Are there no papers for nitrolite to remove ions from aqueous solution? You should add more information. Also, is Nitrolite product name? I think nitrolite is not minerl name.
- Materials and methods 72: “Its conductivity was 0.1 uS/cm.” Why did you measure the conductivity? I can’t understand the relation to this paper.
- Materials and methods: You should add how to adjust the pH of the solution. Also, you should add the method for Cr(VI) experiment of 3.6.
- Results and discussion 93-98: This part moves to “Materials and methods”.
- Figure 1: (a), (b), (c), (d), (e) labels are set at top-left of the graph.
- Results and discussion: The figures are moved to after the paragraph of figure explanation. The order “The paragraph for explanation, results and discussion” →”Figure” is better to understand.
- Results and discussion 108: “the reason for selectivity.” What is selectivity? You should add the sentences for explanation of selectivity.
- Results and discussion page 4 114: There is no description for Cr(III). You should add the results for Cr(III).
- Results and discussion 120-124: “During the ion exchange,,,, at the initial pH 5.” This part is moved to before Figure 2.
- Figure 2,3,4: Plots are difficult to understand. La:, Ce: ▲, Pr:■, Nd: ◆,Cr: ○, are better.
- Results and discussion 142-143: “the values of lanthanides ,,, Medusa.” is moved to l. 134 after Nitroite.
- Table 3: You should improve the table to make it more clear.
- L153-163: This part is moved to before Table 3.
- L226-230: This part is moved to L 165.
- L 245 – 254: This part is moved to L 242 after “in Table 5.”.
- L 259-260: “The desorption ,,, in Figure 5.” is moved to L 257 after “at pH 5.”.
- 8. comparison with the other clinoptlolite: Why did you compare nitrolite with only clinoptilolite? You should compare some natural adsorbents.
I recommended publication of this paper, subject to the above points being satisfactorily addressed.
Author Response
Response to Reviewer 2 Comments
Thank you very much for kind revision. All comments were included in the revised paper. It allowed to improve significantly the paper.
Comment 1
Title: This is very ambiguous. For example, “Sorption behaviors of light lanthanides (III) (La(III), Ce(III), Pr(III), Nd(III)) and Cr(III) using Nitrolite” is better to understand this paper.
Response 1
The title has been changed as recommended
Comment 2
Abstract 9: “light lanthanides (III)”→”light lanthanides (III) (La(III), Ce(III), Pr(III), Nd(III))”. i
Response 2
The record has been changed as recommended
Comment 3
Abstract 10: “The investigated ins were ,,,(ICP-OES).” delete.
Response 3
The sentence has been deleted
Comment 4
Abstract: You should add the isotherm results and indicate which isotherm models and kinetics models are the best.
Response 4
The following sentences have been added to the abstract:
The sorption capacities of La(III) 4.77 mg/g, Ce(III) 4.45 mg/g, Pr(III) 4.30 mg/g, Nd(III) 4.13 mg/g and Cr(III) 2.39 mg/g were calculated from Langmiur model which describe adsorption better than Freundlich and Dubinin-Radushkevich.
The sorption kinetics of investigated ions was described by pseudo-second-order model the best.
Comment 5
Introduction: There are shortage of the information for Nitrolite. Are there no papers for nitrolite to remove ions from aqueous solution? You should add more information. Also, is Nitrolite product name? I think nitrolite is not minerl name.
Response 5
I found two articles. In both the characteristic of Nitrolite is very short: “Nitrolite (a partially processed mineral zeolite)”.
The Nitrolite was used for removal of PO43-, NO3-and NH4+ ions from waste waters.
Both articles have been citted [14, 15].
Yes, Nitrolite is a commercial product name and it is supplied by Purolite International, Ltd. (United Kingdom), as stated in the materials section (2.1).
Comment 6
Materials and methods 72: “Its conductivity was 0.1 uS/cm.” Why did you measure the conductivity? I can’t understand the relation to this paper
Response 6
The conductivity of water was measured to control its quality. This water was used to preparation of metal ion solutions.
Comment 7
Materials and methods: You should add how to adjust the pH of the solution. Also, you should add the method for Cr(VI) experiment of 3.6.
Response 7
The following sentences have been added to the Materials and methods section:
The pH values were adjusted by using sodium hydroxide and nitric acid solutions.
Removal of light lanthanides(III) and chromium(III) from chromium(VI) ions solutions were done at pH value 3.5. 1 g of Nitrolite was equilibrated with 50 mL of the solution containing La(III), Ce(III), Pr(III), Nd(III), Cr(III) and chromium(VI) at concentration of 100 mg/L of each metal ions.
Comment 8
Results and discussion 93-98: This part moves to “Materials and methods”.
Response 8
The lines 93-98 have been moved to “Materials and methods”.
Comment 9
Figure 1: (a), (b), (c), (d), (e) labels are set at top-left of the graph
Response 9
Labels have been set at top-left of the graph
Comment 10
Results and discussion: The figures are moved to after the paragraph of figure explanation. The order “The paragraph for explanation, results and discussion” →”Figure” is better to understand.
Response 10
The figures have been moved to after the paragraphs of their explanation.
Comment 11
Results and discussion 108: “the reason for selectivity.” What is selectivity? You should add the sentences for explanation of selectivity.
Response 11
I meant sorption selectivity of ions. The following sentences have been added.
However, the light lanthanides are so similar, that the differences in sorption, thus selectivity of Nitrolite towards them is low.
Comment 12
Results and discussion page 4 114: There is no description for Cr(III). You should add the results for Cr(III).
Response 12
The following sentences have been added:
The sorption capacities of chromium(III) ions increase with increasing pH and are respectively: 0.6 mg/g at pH 2, 1.9 mg/g at pH 3.5 and 2.4 mg/g at pH 5.
Comment 13
Results and discussion 120-124: “During the ion exchange,,,, at the initial pH 5.” This part is moved to before Figure 2.
Response 13
The line 120-124 have been moved to before Figure 2
Comment 14
Figure 2,3,4: Plots are difficult to understand. La:, Ce: ▲, Pr:■, Nd: ◆,Cr: ○, are better.
Response 14
Figures 2,3,4 have been changed according to comment.
Comment 15
Results and discussion 142-143: “the values of lanthanides ,,, Medusa.” is moved to l. 134 after Nitroite.
Response 15
The lines 142 -143 have been moved to line 134
Comment 16
Table 3: You should improve the table to make it more clear.
Response 16
The Table 3 has been improved to make it more clear.
Comment 17
L153-163: This part is moved to before Table 3.
Response 17
The lines 153-163: This part has been moved to before Table 3.
Comment 18
L226-230: This part is moved to L 165.
Response 18
The lines 226-230 have been moved to line 165.
Comment 19
L 245 – 254: This part is moved to L 242 after “in Table 5.”.
Response 19
The part from line 245-254 has been moved to line 242 after “in Table 5.”.
Comment 20
L 259-260: “The desorption ,,, in Figure 5.” is moved to L 257 after “at pH 5.”.
Response 20
The Figure 5 has been deleted according the Reviewer 1 comment.
Comment 21
- comparison with the other clinoptlolite: Why did you compare nitrolite with only clinoptilolite? You should compare some natural adsorbents.
Response 21
Thank you for this comment. The subsection 8 has been changed on Comparison with the other natural sorbents. The Nitrolite has been compared to some natural adsorbents like Kaolinite, Smectite and Montmorillonite.
Natural sorbent Kaolinite was used for removal of Nd ions. At initial concentration 42.3 µmol/L of neodymium the capacity was 0.576 mg/g and was lower than for Nitrolite [31]. Smectite was used for sorption of lanthanides. This natural sorbent is able to adsorb light lanthanides similarly to Nitrolite in the pH range 3-8 [32]. The natural sorbents can be modified to improve the sorption capacity. The Montmorillonite was modified by hexadecyl trimethyl ammonium bromide (HDTMA) and sodium diethyl dithiocarbamate (DDTC). The maximum of sorption capacity for La(III) was 0.178mmol/g and was higher than for Nitrolite [33].
